# Genome-Wide In Silico Identification, Structural Analysis, Promoter Analysis, and Expression Profiling of *PHT* Gene Family in Sugarcane Root under Salinity Stress

Naveenarani Murugan, Vignesh Palanisamy, Mahadevaiah Channappa, Valarmathi Ramanathan, Manimekalai Ramaswamy, Hemaprabha Govindakurup and Appunu Chinnaswamy *

Division of Crop Improvement, Indian Council of Agricultural Research-Sugarcane Breeding Institute, Coimbatore 641007, Tamil Nadu, India
* Correspondence: cappunu@gmail.com or c.appunu@icar.gov.in

**Abstract:** The phosphate transporter (PHT) family of proteins plays an imperative role in regulating phosphorus (P) acquisition as well as in translocation from the soil into cells and organs. Phosphorus is an essential macronutrient required for plant life that is not readily available to crops, and resources are diminishing rapidly because of the huge needs of global agriculture. In this study, 23 *ShPHT* genes were identified in the sugarcane (*Saccharum* spp.) genome through a comprehensive genome-wide in silico analysis. Phylogeny, gene structure, and conserved motif analysis of *PHT* genes in sugarcane (*ShPHTs*) indicated five subfamilies (PHT1-4 and PHO1 subfamily). Gene ontology (GO) analysis revealed that the *ShPHT* genes were largely involved in phosphate ion transport, phosphate starvation, stimulus response, stress response, and symporter activity. Gene expression analysis under salinity stress confirmed strong induction of *PHT* genes in wild genotype sugarcane (IND99-907). *PHT1-1*, *PHT1-2*, and *PHT1-3* members were notably up-regulated in roots under salt stress. The upstream region of *PHT* genes contained PHR1-binding sites (P1BS), MYB-type, and WRKY-type binding elements. Overall, the present study paves the way for a deeper understanding of the evolution of sugarcane *PHT* genes and their role in salinity and Pi stress tolerance in sugarcane.

**Keywords:** sugarcane; phosphate transporter (PHT); genome-wide analysis; in silico; 3D structure; salinity stress; cis-element prediction; promoter; transient expression

## 1. Introduction

Phosphorus (P) is the second major macronutrient involved in molecular regulations such as photosynthesis, energy metabolism, signal transduction, and enzymatic regulation, and plays a crucial role in plant growth and development. In soil, phosphate is present in two forms, namely, organic (20–80%) and inorganic (<5%). Although it is rich in the environment, its level of distribution and availability as a soluble form in the soil is much less. Therefore, the application of P fertilizers has increased globally from 4.6 million tons in 1961 to 50 million tons in recent years [1,2]. Frequent applications and excess use of P fertilizer have led to the depletion of non-renewable rock phosphate as well as other environmental concerns. Predominantly, plants can utilize only the inorganic phosphate from the soil as a phosphate source, which is readily absorbed by plant roots through phosphate (Pi) transporters and transported via various strategies to enhance Pi uptake from P-limited soils by remodeling the root morphology, secreting organic acids and phosphatases, and inducing the high-affinity Pi transporters [3]. Pi uptake in roots mainly depends on soil characteristics, which are highly influenced by abiotic stresses, organic compounds, and concentration of metals such as Fe, Al, and Ca [4]. The phosphate transporter (*PHT*) transporter gene family in plants plays an important role in Pi acquisition and translocation in plants; their expression varies depending on their high affinity or low affinity for phosphate.

*PHT* genes are categorized into six groups, namely, PHT1, PHT2, PHT3, PHT4, PHT5, and PHO1, based on evolutionary relationships, protein sequences, structures, and functions [5,6]. In recent years, with increased efforts around whole-genome sequencing, all the members of the *PHT* gene family have been reported in a variety of species [7]. Members of the Pi transporter family are mostly expressed in roots, and others in shoots, leaves, and flowers [8]. PHT1, the high-affinity Pi transporters, mainly expressed in the plasma membrane of root cells, play a vital role in both Pi uptake from the soil and translocation [7]. The Pi transporter subfamilies, namely, PHT2/PHT4, PHT3, and PHT4, localize between the cytoplasm and plastids, mitochondria, and Golgi membranes for energy metabolism, respectively. The PHT5 family plays a crucial role in vacuolar mediating Pi storage and adaptation [9]. PHO1 has been reported in both Pi transport from root to shoot and Pi-deficiency response signal transduction cascade [10].

Abiotic stresses, such as drought, salinity, extreme high and low temperatures, and heavy metals have a devastating impact on plant growth and development, and result in a worldwide reduction in crop productivity. Salinity has been considered the most serious abiotic stress factor in plants, leading to severe dehydration and inorganic solute imbalance in major crop species, ultimately leading to limited growth and yield. Drought and salinity have a strong impact on plant nutrient relations as well. It has been reported that drought stress often diminishes N and P concentrations in plant tissues [11]. Due to salinity, the phosphate ions in the soil tend to form an insoluble phosphate that limits P availability, absorption, and transport from the soil by root to shoot. Therefore, combined exposure to salinity stress and Pi starvation is deleterious for plant growth [12]. During abiotic and biotic stresses, plants tend to regulate the expression of a particular gene to adapt to the adverse effects, and the expression of the specific gene is controlled by the upstream cis-acting elements, which are key links in plant stress responses. Previous reports have shown that the majority of Pi transporters contain putative cis-acting regulatory elements in their promoters associated with plant stress and defense signaling [13].

Sugarcane (*Saccharum* spp. hybrid) is cultivated worldwide in tropical and subtropical regions as a major source of sugar and a viable source of biomass for ethanol production. At present, various stress factors restrict yields to up to 50% in major crops. In sugarcane, salinity is one of the major abiotic stresses that adversely affects the growth rate at various developmental stages [14,15]. On the other hand, due to phosphate-deficient soils, the use of P fertilizer in sugarcane plantations has increased year over year to improve crop growth and development, leaving a large P footprint. Therefore, it is important to develop salt-tolerant plants that facilitate or even boost the availability and absorption of soil phosphorus more efficiently under these conditions [16]. In this study, the sugarcane *PHT* gene family members are investigated using various bioinformatics tools. The molecular characteristics, phylogenetic relationships, gene structures, conserved domains, evolutionary relationships, and cis-elements are systematically studied. To date, while studies have been conducted in different plants, many structural aspects and ligand-binding aspects of the PHT protein families remain unknown. Hence, this study aims at homology modeling of PHT proteins, post-translational modifications, and functional roles of PHT genes through expression profiles between salt-tolerant and salt-sensitive sugarcane varieties. In addition, this study provides insights for future research on *ShPHT* genes associated with growth, development, and stress tolerance in sugarcane.

## 2. Materials and Methods

### 2.1. Data Retrieval and Identification of Sugarcane PHT Genes

The sequences of *PHT* genes of *Arabidopsis thaliana* (TAIR10), *Zea mays* (Zm-B73-REFERENCE-NAM-5.0), and *Sorghum bicolor* (v3.1.12) were retrieved from the Ensembl Plants database (http://plants.ensembl.org/, (accessed on 12 May2022)) and the Sugarcane genome of *Saccharum* spp. hybrid (Sh) cultivar R570 was retrieved from Sugarcane Genome Hub (https://sugarcane-genome.cirad.fr/, (accessed on 12 May 2022)). To find the potential orthologous genes encoding PHTs in Sugarcane, the protein sequences of *A. thaliana*, *Z.*

*mays*, and *S. bicolor* were used as queries in BLAST with E-10 as the threshold. The protein, gene, CDS, and gff3 annotation files were downloaded for further analysis. Transcripts encoding less than 90% similarity and duplicated transcripts were removed to generate the final dataset. The identified sugarcane PHT proteins were designated using the species abbreviation, the gene family name, and their members, for example, ShPHT1-1.

### 2.2. Sequence Analysis of PHT Proteins from Sugarcane

The physiochemical characteristics, including molecular weights (MW), theoretical iso-electric point (pIs), instability index, grand average of hydropathicity (GRAVY), and protein lengths (aa) of the candidate ShPHT protein sequences, were acquired using the ExPASy—ProtParam tool (http://web.expasy.org/protparam/, (accessed on 17 May 2022)) [17]. TMHMM server v.2.0 (http://www.cbs.dtu.dk/services/TMHMM/, (accessed on 17 May 2022)) [18] was used to predict the transmembrane domains (TMD). Potential phosphorylation sites were identified using NetPhos 3.1 server (http://www.cbs.dtu.dk/services/NetPhos/, (accessed on 19 May 2022)) [19] with a potential value > 0.5, and NetNGlyc 1.0 server (http://www.cbs.dtu.dk/services/NetNGlyc/, (accessed on 19 May 2022)) [20] was used to predict the N-glycosylation sites in the amino acid sequence of phosphate transporter proteins. The subcellular localization of ShPHTs were predicted using the WoLF-PSORT tool (http://www.genscript.com/wolf-psort.html, (accessed on 26 May 2022)) [21].

### 2.3. Phylogenetic Analysis of PHT Genes

To explore the phylogenetic relationship of *PHT* genes between *Saccharum* spp. hybrid, *A. thaliana*, *Z. mays*, and *S. bicolor*, a multiple sequence alignment was performed for the candidate phosphate transporter proteins using CLC Genomics Workbench with default parameters (Qiagen, Hilden, Germany). The results were used to construct a phylogenetic tree by the Neighbor Joining (NJ) method with pair-wise deletion, Poisson correction, and 1000 bootstrap replicate parameters, then visualized using MEGA-X software [22].

### 2.4. Gene Structure Analysis and Identification of Conserved Motifs

The intron/exon organization was established by aligning the genomic sequences with the CDS sequences. Gene Structure Display Server 2.0 (GSDS 2.0, http://gsds.cbi.pku.edu.cn/, (accessed on 20 May 2022)) [23] was used to generate a schematic diagram of the exon–intron organization of *ShPHT* genes. To verify the presence of conserved motifs, the protein sequences were submitted to the NCBI conserved domain database (http://www.ncbi.nlm.nih.gov/Structure/cdd/wrpsb.cgi, (accessed on 17 May 2022)) [24], SMART (http://smart.embl-heidelberg.de/, (accessed on 25 May 2022)) [25], and Pfam server (http://pfam.xfam.org/, (accessed on 25 May 2022)) [26], and the results were visualized using TBtools (https://github.com/CJ-Chen/TBtools, (accessed on 28 May 2022)) [27]. To understand the occurrence of motif diversity and conservation, the protein sequences of ShPHTs were analyzed using MEME Suite version 5.3.3 (https://meme-suite.org/meme/, (accessed on 23 May 2022)) using the default parameters and with the maximum number set to 20 [28]. The MEME results were then visualized using TBtools software.

### 2.5. Prediction of Structure, Protein Pocket Sites, and Protein Modeling

The protein secondary structures of the ShPHTs were predicted using the SOPMA online server [29]. As the crystal structures of most of the PHT proteins are not available in the Protein Data Bank (PDB), homology-based structural modelling of ShPHT proteins was predicted using the Phyre2 web server [30]. The CASTp (Computed Atlas of Surface Topography of proteins) tool (http://sts.bioe.uic.edu/castp/calculation.html, (accessed on 31 May 2022)) was used to predict the active site pockets and topology of the ShPHT proteins [31]. Model quality was checked by Ramachandran plot analysis in VADAR (Volume, Area, Dihedral Angle Reporter) server [32].

### 2.6. GO and KEGG Analysis

Gene ontology (GO) analysis was carried out using the PANNZER2 web server (http://ekhidna2.biocenter.helsinki.fi/sanspanz, (accessed on 20 May 2022)). *ShPHT* genes were analyzed for their GO functions [33]. KEGG analysis using the BlastKOALA (KEGG Orthology and Links Annotation) web server (https://www.kegg.jp/blastkoala, (accessed on 20 May 2022)) [34] was employed to characterize the individual functions of the genes.

### 2.7. Chromosomal Organization and Ka/Ks Calculation

The chromosomal localization information of *ShPHT* genes was analyzed using the MapGene2chromosome web v2.1 (MG2C) database (http://mg2c.iask.in/mg2c_v2.1/, (accessed on 28 May 2022)) according to their position information, available on the Sugarcane Genome Hub (https://sugarcane-genome.cirad.fr/, (accessed on 20 May 2022)). The nonsynonymous substitution rate (Ka), synonymous substitution rate (Ks), and Ka/Ks values were calculated using the PAL2NAL program with default settings [35]. We calculated the divergence time of *ShPHT* genes with the formula $T = Ks/ (2 \times 6.1 \times 10^{-9}) \times 10^{-6}$ Mya [36].

### 2.8. Expression Study of PHT Genes in Sugarcane

The RNA-Seq dataset generated earlier by our group was used for the expression profiling of *PHT* gene families under salinity stress in sugarcane species. The salinity stress dataset is available in the NCBI database under BioProject accession number PRJNA716503 [37]. In the present study, expression analysis of *PHT* genes was performed between stress and control root samples of *E. arundinaceus*, a wild relative of sugarcane (IND 99-907), and the salinity-sensitive genotype Co 97010. De novo assemblies for IND 99-907 and Co 97010 were generated separately, and unigenes were clustered using CD-HIT with a word size of 8 and a percent similarity threshold of 99 percent. The expression values (FPKM) were obtained by mapping the cleaned reads against the individual assemblies containing unigenes using the RSEM tool. RNA isolation and cDNA synthesis were carried out as previously described [37]. IDT was used to create the primers. The internal control was the Glyceraldehyde 3-phosphate dehydrogenase (GAPDH) gene. For quantitative real-time PCR (qRT-PCR) analysis, a StepOne real-time PCR system (Applied Biosystems, Burlington, ON, Canada) was used with the following temperature profile: 10 min of denaturation at 95 °C, followed by 40 cycles 15 s of denaturation at 95 °C, 1 min of annealing, and extension at 60 °C in a final volume of 25 μL reaction. The relative expression of *PHT* genes in IND 99-907 and Co 97010 was determined using $2^{-\Delta\Delta Ct}$ method [38]. Three biological and technical replicates were employed for expression analysis. The heatmap for the expression data was generated using TBtools.

### 2.9. Prediction of Putative cis-Regulatory Elements and Transient Expression Analysis of EaPHT1:2 Promoter

To predict the cis-regulatory elements of *ShPHT* genes, we obtained the region 2000 bp upstream from the transcription start site (TSSs) of the genome using the faidx option from the tool samtools. The PLACE database (http://www.dna.affrc.go.jp/htdocs/PLACE/, (accessed on 2 June 2022)) was used to predict the cis-elements in the promoter region [39]. The 5′ regulatory promoter region of the *EaPHT1-2* gene was isolated using the RAGE technique (data not included), and transient expression analysis was carried out as described in [40].

## 3. Results

### 3.1. Identification and Characterization of PHT Genes in Sugarcane

A total of 23 putative *PHT* genes were identified in the sugarcane genome (*Saccharum* spp. hybrid (Sh) cultivar R570) (Table 1) using keyword search and BLASTP search against the *PHT* gene families of the *Arabidopsis thaliana* (TAIR10), *Zea mays* (Zm-B73-REFERENCE-NAM-5.0), and *Sorghum bicolor* (v3.1.12) genomes as a background database.

Based on conserved domain analysis, the highly conserved domains of PHT proteins were MFS_1 (Major Facilitator Superfamily), the sugar transport domain in PHT1, the PHO4 (PHOSPHATE4) domain in PHT2, the mito_carr domains in PHT3, MFS_1, and MFS_3, the sugar_tr domains in PHT4, and the EXS superfamily, SPX, and SPX superfamily domains in PHO (Figure 1). Physio-chemical and biochemical analyses of the ShPHT proteins were characterized using ProtParam, and the results are presented in Table 1. We found that the *Saccharum* spp. hybrid has mostly positively charged amino acids in ShPHT proteins ranging from 216 to 877 aa, with an estimated molecular weight of 23.926 to 99.766 kDa. The instability index of ShPHT proteins varied from 23.1 to 48.94, while their aliphatic index ranged from 84.38 to 107.9. Furthermore, the ShPHT proteins' GRAVY ranged from −0.137 to 0.707, and their pI (theoretical isoelectric point) ranged from 7.13 to 10.04, showing that most proteins are alkaline, with one protein ShPHT4-2 being slightly acidic (6.76).

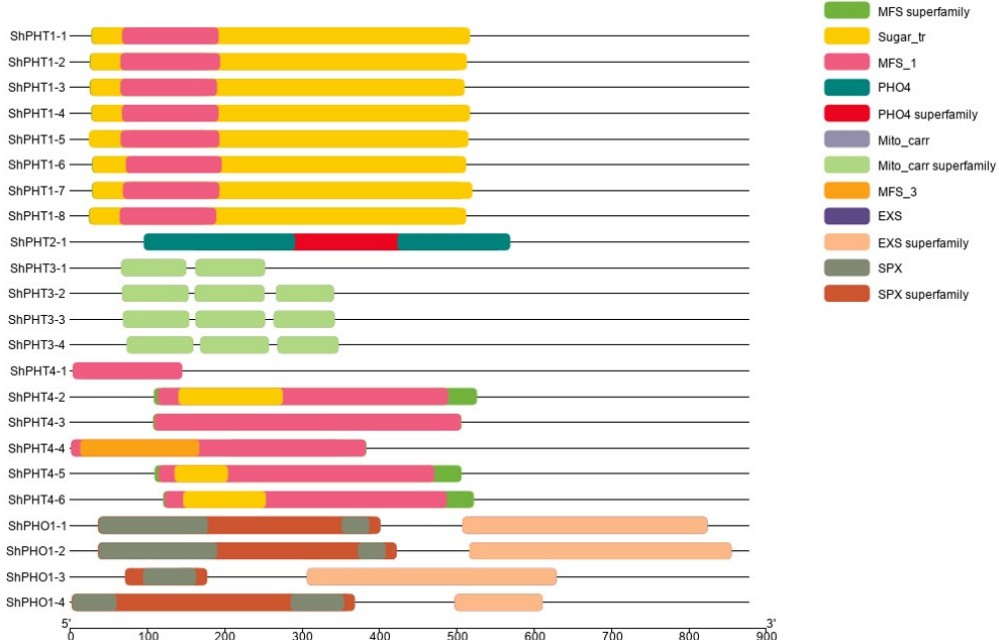

**Figure 1.** Conserved domains in ShPHT proteins were identified using the NCBI-CD database. The domains are represented by different colors. Schematic representation of the conserved domains integrated employing TBtools.

### 3.2. Phylogenetic Analysis

To understand the evolutionary relationship of *PHT* family genes, the amino acid sequences of 18 *A. thaliana*, 24 *Z. mays*, 27 *S. bicolor*, and 23 *Saccharum* spp. hybrid members were used to construct an unrooted neighbor-joining (NJ) tree. Through phylogenetic analysis, we discovered that the *PHT* gene family is divided into five clades (Figure 2). Clade I contains the most members, with 42 PHT proteins, 13 ZmPHT, 12 SbPHT, 9 AtPHT, and 8 ShPHT (ShPHT1-1 to ShPHT1-8) members. Clade II contains 13 genes. All genes in the PHT4 subfamily occupied Clade IV. Clades I and II both had 100% bootstrap support, and their members are highly conserved. Clades III, IV, and V consist of members of PHT3, PHT2, and PHO1. It is interesting to note that the proteins of PHT1 are closely related among monocot species rather than *A. thaliana*, suggesting that they diverged from a common ancestor.

**Table 1.** List of the identified *PHT* genes and their characteristics in the *Saccharum* spp. hybrid genome.

| Gene Name | Gene ID | Genomic Position (bp) | Strand | CDS (bp) | Protein Length (A.A) | Protein Molecular Weight (kDa) | Theoretical pI | Number of Negatively Amino Acid | Number of Positively Amino Acid | Instability Index | Aliphatic Index | GRAVY | Trans Membrane Domain |
|---|---|---|---|---|---|---|---|---|---|---|---|---|---|
| ShPHT1-1 | Sh_251L03_g000070 | Sh01:30199785–30202295 | + | 1626 | 541 | 58.814 | 7.63 | 40 | 41 | 30.26 | 88.82 | 0.349 | 11 |
| ShPHT1-2 | Sh_251L03_g000050 | Sh01:30228262–30231839 | - | 1575 | 524 | 57.087 | 7.13 | 37 | 37 | 31.84 | 90.97 | 0.368 | 11 |
| ShPHT1-3 | Sh_236J19_g000050 | Sh01:67384705–67386273 | - | 1569 | 522 | 57.224 | 8.92 | 36 | 43 | 27.01 | 94.62 | 0.409 | 11 |
| ShPHT1-4 | Sh_229B19_p000040 | Sh01: 67384705–67386273 | - | 1578 | 525 | 57.512 | 8.7 | 38 | 43 | 28.67 | 93.33 | 0.385 | 11 |
| ShPHT1-5 | Sh_216E19_g000030 | Sh02:12496129–12497748 | - | 1620 | 539 | 58.89 | 8.66 | 47 | 52 | 36.71 | 89.98 | 0.294 | 12 |
| ShPHT1-6 | Sh_247G22_g000050 | Sh03:29909878–29911962 | - | 1656 | 551 | 60.285 | 8.6 | 44 | 48 | 33.2 | 93.9 | 0.336 | 12 |
| ShPHT1-7 | Sh_222J11_g000100 | Sh01:69058018–69059604 | + | 1587 | 528 | 56.698 | 8.1 | 33 | 35 | 38.63 | 89.15 | 0.396 | 11 |
| ShPHT1-8 | Sh_222A01_g000030 | Sh06:3102321–3103931 | - | 1611 | 536 | 58.81 | 8.3 | 40 | 43 | 34.94 | 84.38 | 0.289 | 12 |
| ShPHT2-1 | Sh_232L11_g000050 | Sh04:24145076–24148283 | + | 1719 | 572 | 59.251 | 9.42 | 24 | 36 | 32.21 | 106.54 | 0.707 | 13 |
| ShPHT3-1 | Sh_252F06_g000020 | Sh01:57191013–57193003 | - | 1044 | 347 | 37.76 | 8.94 | 26 | 35 | 23.1 | 92.85 | 0.283 | 0 |
| ShPHT3-2 | Sh_208E04_contig-2_g000020 | Sh04:39635010–39639407 | - | 1119 | 372 | 39.135 | 9.23 | 24 | 35 | 38.3 | 88.23 | 0.259 | 0 |
| ShPHT3-3 | Sh_234H01_g000060 | Sh10:11858441–11864575 | + | 1113 | 370 | 39.113 | 9.22 | 23 | 34 | 44.05 | 86.38 | 0.268 | 0 |
| ShPHT3-4 | Sh_247B01_g000110 | Sh06:11674421–11678619 | + | 1104 | 367 | 38.412 | 9.39 | 20 | 34 | 37.81 | 84.99 | 0.292 | 0 |
| ShPHT4-1 | Sh_227B04_g000030 | Sh01:21002885–21004607 | + | 651 | 216 | 23.926 | 9.73 | 12 | 22 | 39.26 | 105.6 | 0.462 | 5 |
| ShPHT4-2 | Sh_241M07_g000080 | Sh02:36755180–36775918 | - | 1623 | 540 | 57.755 | 6.76 | 38 | 37 | 44.48 | 102.94 | 0.377 | 9 |
| ShPHT4-3 | Sh_227N18_g000090 | Sh03:15436854–15440720 | + | 1542 | 513 | 55.383 | 8.42 | 31 | 34 | 45.05 | 97.31 | 0.346 | 9 |
| ShPHT4-4 | Sh_227D07_g000010 | Sh03:21537139–21541012 | - | 1173 | 390 | 42.392 | 9.27 | 17 | 25 | 48.94 | 107.9 | 0.566 | 9 |
| ShPHT4-5 | Sh_220N05_g000170 | Sh03:44310739–44314195 | - | 1554 | 517 | 55.814 | 10.04 | 19 | 38 | 42.72 | 100.43 | 0.475 | 10 |
| ShPHT4-6 | Sh_217I09_g000040 | Sh09:24432214–24435720 | + | 1590 | 529 | 55.44 | 9.65 | 19 | 35 | 37.44 | 97.2 | 0.615 | 11 |
| ShPHO1-1 | Sh_244C06_g000040 | Sh10:19140506–19145537 | - | 2541 | 846 | 96.534 | 8.65 | 90 | 99 | 41.61 | 89.78 | −0.137 | 6 |
| ShPHO1-2 | Sh_226O09_contig-2_g000020 | Sh10:19140506–19145537 | - | 2634 | 877 | 99.766 | 8.44 | 94 | 101 | 42.78 | 87.5 | −0.162 | 5 |
| ShPHO1-3 | Sh_254O18_g000050 | Sh03:11344984–11348425 | - | 1956 | 651 | 73.3 | 9.1 | 54 | 69 | 41.96 | 92.44 | −0.056 | 5 |
| ShPHO1-4 | Sh_253H03_g000040 | Sh03:11346722–11350230 | - | 1908 | 635 | 70.213 | 8.82 | 65 | 74 | 42.67 | 85.64 | −0.154 | 4 |

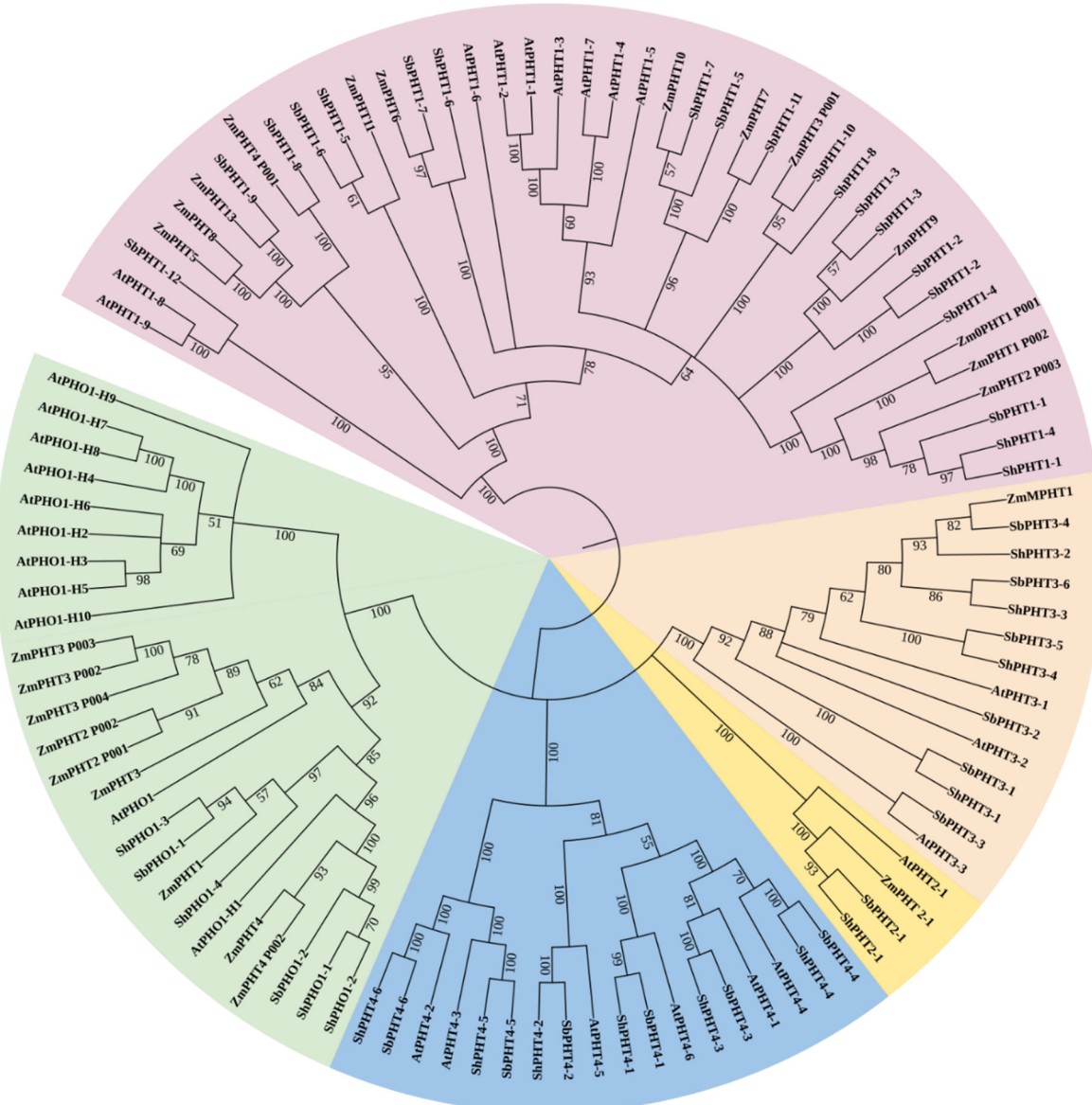

**Figure 2.** Phylogenetic analysis of PHT proteins between *Saccharum* spp. hybrid, *Sorghum bicolor*, *Zea mays*, and *Arabidopsis thaliana*. An unrooted phylogenetic tree was constructed by MEGA X using the neighbour-joining method with 1000 bootstrap replicates. Five clades were identified, and are represented here as Clade I—Violet, Clade II—Yellow, Clade III—Orange, Clade IV—Blue, and Clade V—Green.

### 3.3. Chromosomal Localization and Duplication of ShPHT Genes

The chromosome locations of 23 *ShPHT* genes were mapped on their corresponding chromosomes using the MapGene2chromosome webserver (Figure 3). The results showed that the maximum number of *ShPHT* genes are localized on chromosome 1 (seven genes) and chromosome 3 (six genes). Out of the seven genes on chromosome 1, five belong to the *PHT*1 subfamily. To investigate the evolution process among the *ShPHT* genes, we computed the evolutionary rates and selective pressure utilizing the Ka (non-synonymous), Ks (synonymous), and Ka/Ks (non-synonymous/synonymous) ratio (ω). A total of seven pairs of duplicated gene clusters (Table 2) were identified and were linked on chromosome 1 (*ShPHT1-1*/*ShPHT1-4*; *ShPHT1-2*/*ShPHT1-3*), chromosome 3 (*ShPHO1-3*/*ShPHO1-4*; *ShPHT4-3*/*ShPHT4-4*), and chromosome 10 (*ShPHO1-1*/*ShPHO1-2*). The Ka/Ks ratios of the gene pairs ranged between 0.037 and 0.3651 and were <0.5, showing that their

development is strongly influenced by purifying selection. The estimated divergence time between the duplication of the gene pairs showed that *ShPHT* in *Saccharum* spp. hybrid cultivar R570 occurred approximately between 1.1 Mya and 6.53 Mya (million years ago), with an average age of 1.13 Mya (Table 2).

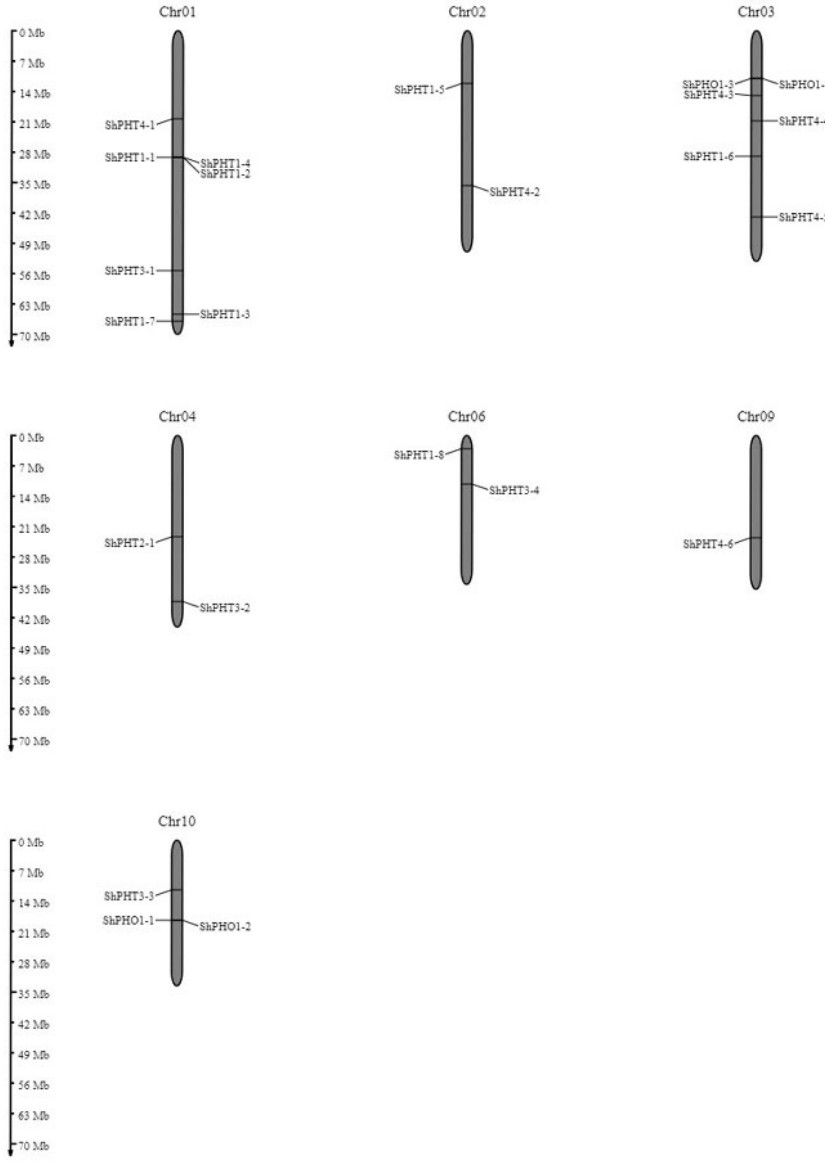

**Figure 3.** The distribution of *ShPHT* genes on the *Saccharum* spp. hybrid R570 genome chromosome scaffolds. MG2C v.2 (http://mg2c.iask.in/mg2c_v2.0/, (accessed on 28 May 2022)) was used to visualize the gene locations on chromosomes.

*3.4. Gene Structure and Conserved Motif Analysis*

To acquire insights into the structural evolution of the *ShPHT* gene family, exon/intron structure was derived based on their coding and genomic regions (Figure 4A). The results revealed that the number of exons in the *ShPHT* gene varied substantially from 1 to 14 exons. Excluding *ShPHT1-6*, seven genes in *ShPHT1* contain one exon, whereas one gene, *ShPHO1-1*, has 14 exons. In particular, the *ShPHO1* and *ShPHT4* subfamily genes contain more introns. This finding indicates a high degree of divergence among the Sh*PHT* genes. To reveal the conserved motifs of *ShPHT* genes more comprehensively, the MEME web server was utilized to estimate the pattern of conserved motifs. The results showed that 20 putative conserved motifs were identified in the ShPHT proteins (Figure 4B), and the conserved

motifs ranged from 21 to 50 amino acids in length (Table S1). Each ShPHT protein had between one and ten conserved motifs; however, two genes *ShPHT2-1* and *ShPHT4-1* were not identified as containing any motifs. The vast majority of members of ShPHT1-1, ShPHT1-2, ShPHT1-4, ShPHT1-5, ShPHT1-6, ShPHT1-7, and ShPHT1-8 contained mostly motif 10. However, ShPHT4-2, ShPHT4-5, and ShPHT4-6 contained only one conserved motif 20. ShPHT4-3 and ShPHT4-4 contained two conserved motifs 19 and 20. Motif 13 and motif 20 were commonly distributed in most of the members of the ShPHT3, ShPHT4, and ShPHO1 families, resulting in the motifs being conserved domain sequences.

*3.5. Structure and Protein Pocket Sites Prediction*

The secondary structure prediction showed that the main components of ShPHTs are the alpha helix (46.93%) and the random coil (15.54%). The presence of extended strand ranges from 8.82 to 20.37%, followed by beta-turn ranging from 2.72 to 7.75% (Table S2). The ShPHT proteins' three-dimensional structures were generated using the Phyre2 web server. For structural prediction, models with high confidence and identity percentage were chosen. All of the 3D protein models were built with 99.8 to 100 percent confidence, and the residue coverage ranged from 72 to 98 percent. The molecular binding pockets vital for protein interaction were detected using the CASTp 3.0 server. The active catalytic sites involved in ligand binding are highlighted as red in the ShPHT structures (Figure 5). The amino acid residues arginine (ARG), alanine (ALA), leucine (LEU), serine (SER), valine (VAL), glycine (GLY), phenylalanine (PHE), glycine (GLY), threonine (THR), isoleucine (ILE), and cysteine (CYS) were the more functionally substantial residues localized in this region. The quality of the projected 3D models was assessed using the VADAR server and Ramachandran Plot analysis, which demonstrated that the residues in the core, favored, and generous regions surpassed 95%, indicating the quality and reliability of the protein structure.

*3.6. Post-Translational Modifications, Transmembrane Domains (TMD), and Subcellular Localization of ShPHT Proteins*

The post-translational modification analysis of ShPHT proteins was carried out in terms of phosphorylation and glycosylation. ShPHT proteins have 19 to 72 sites for potential phosphorylation modifications (Figure S2). Most of the phosphorylation events were predicted to be related to serine (554), followed by threonine (330) and then by tyrosine (100). The proteins ShPHO1-2 and ShPHO1-1 showed a maximum of 72 and 71 potential phosphorylation sites. In ShPHT4-1 and ShPHT3-1 protein, 19 and 23 sites were predicted, whereas in other proteins phosphorylation events ranged from 28 to 66 sites (Table S3). In addition, the N-linked glycosylation pattern on ShPHT proteins was analyzed; the results predicted a total of 41 N-glycosylation sites (Figure S2). Among these N-glycosylation sites, seven amino acid residues were NSTT, four residues were NSSD, and three residues were NVSA, two (NSTV, NLTE, NNST, NSTG, NYTF, NSSS, NKTK) and one (NITR, NLTQ, NLTL, NQTG, NSTV, NKTK), respectively. Except for ShPHT2-1, ShPHT3 members, ShPHT4-1, and ShPHT4-6, all other ShPHTs showed at least one N-glycosylation site. The proteins ShPHO1-1 and ShPHO1-2 were predicted to have five glycosylation sites, while one glycosylation site each was predicted in ShPHT1-5, ShPHT4-3, and ShPHT4-4 (Table S3). Transmembrane domains in ShPHTs were predicted, and ranged from 0 to 13. ShPHT2-1 had the most TMDs, while the members of the ShPHT3 genes had no potential TMDs. Meanwhile, the ShPHT1 proteins showed that most TMDs with 11 or 12 transmembrane domains; ShPHT4 and ShPHO1 had 5-11 and 4-6 TMDs, respectively (Table 1). Subcellular location analysis showed that most ShPHT1, ShPHO1-3, and ShPHO1-4 are localized in the plasma membrane, while the ShPHT2-1 and ShPHT3-4 proteins are localized in the chloroplast and ShPHT4-5 is localized in the chloroplast/mitochondria (Figure S1).

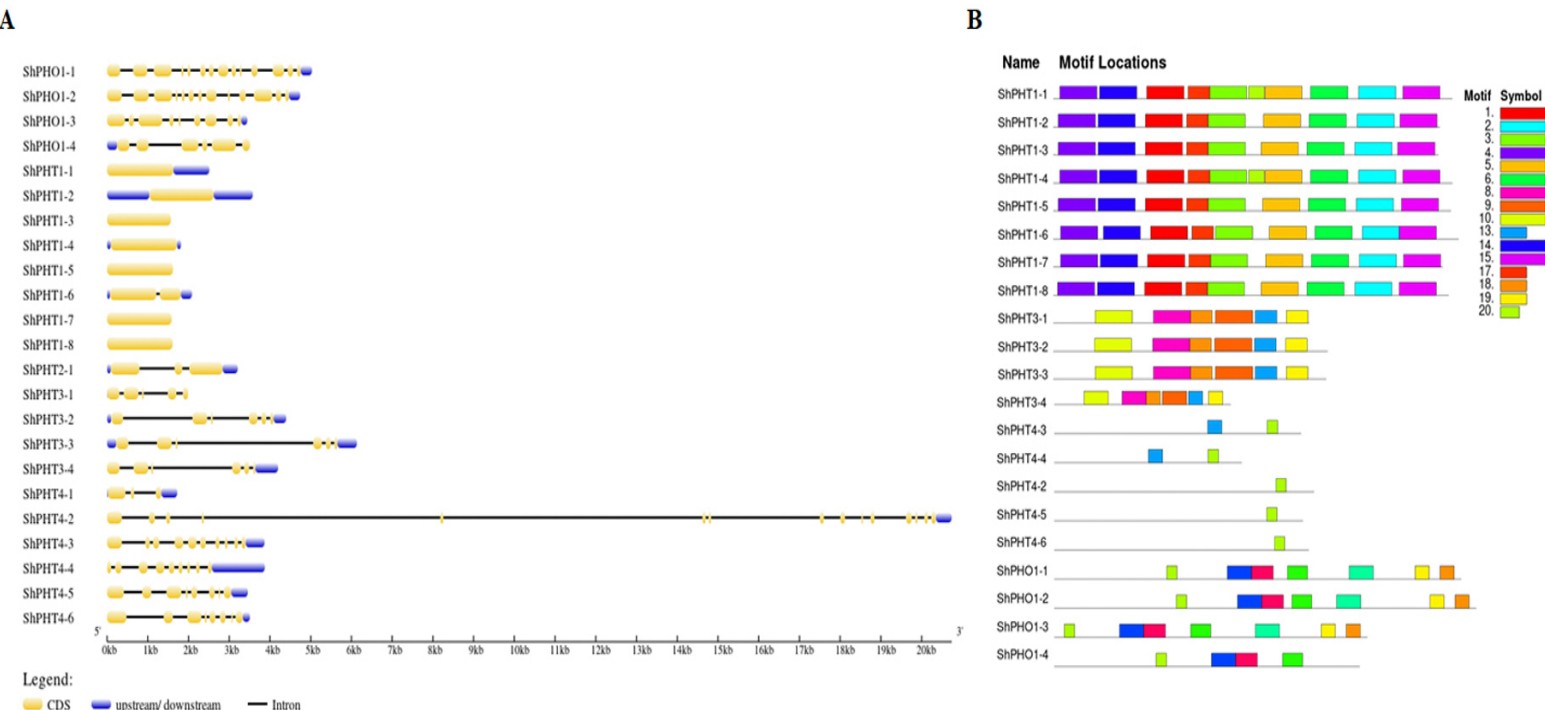

**Figure 4.** Schematic representation of the intron–exon structure and conserved motif analysis of *PHT* genes in Sugarcane. (**A**) The intron–exon structure of the *ShPHT* genes was displayed using an online GSDS tool. Yellow boxes represent exons (CDS), black lines represent introns, and light blue boxes represent 5′ and 3′ untranslated regions. (**B**) The distribution of putative conserved motifs of PHT proteins was mined in the MEME program. Twenty motifs are indicated by different colored boxes and the regular motif sequences are shown in Table S1. The MEME Motif pattern was visualized using TBtools.

**Table 2.** Duplicated gene pairs and the Ka/Ks values predicted in *ShPHT* genes.

| Gene Name | Orthologs | Ka | Ks | Ka/Ks | Divergence Time (Mya) |
|---|---|---|---|---|---|
| ShPHT1-1 | ShPHT1-4 | 0.0027 | 0.0552 | 0.0491 | $4.50 \times 10^{12}$ |
| ShPHT1-2 | ShPHT1-3 | 0.0864 | 1.4853 | 0.0582 | $1.22 \times 10^{14}$ |
| ShPHT3-2 | ShPHT3-3 | 0.0875 | 0.7968 | 0.1098 | $6.53 \times 10^{13}$ |
| ShPHT4-3 | ShPHT4-4 | 0.1088 | 2.9394 | 0.037 | $2.41 \times 10^{14}$ |
| ShPHT4-5 | ShPHT4-6 | 0.4846 | 4.1984 | 0.1154 | $3.44 \times 10^{14}$ |
| ShPHO1-1 | ShPHO1-2 | 0.0029 | 0.0154 | 0.1904 | $1.26 \times 10^{12}$ |
| ShPHO1-3 | ShPHO1-4 | 0.0489 | 0.134 | 0.3651 | $1.10 \times 10^{13}$ |

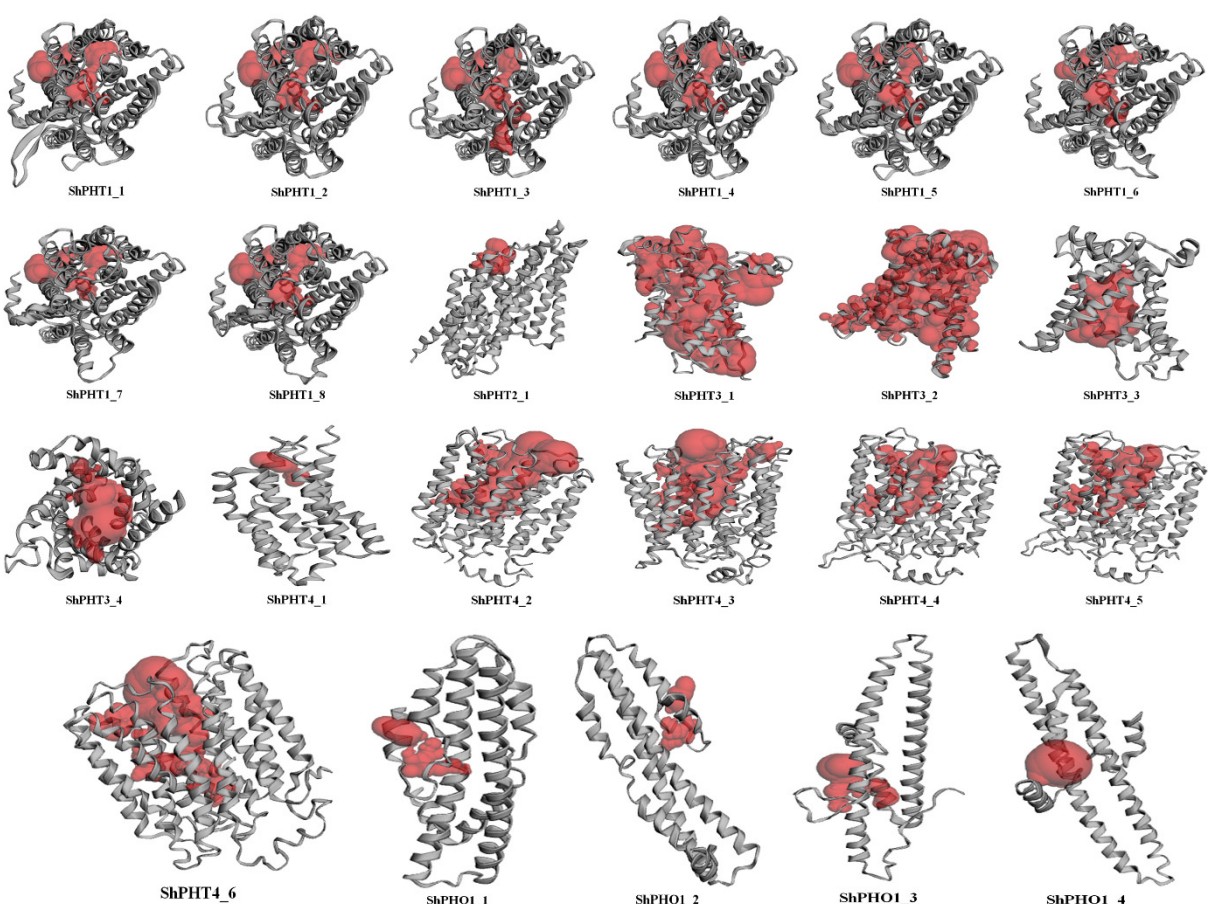

**Figure 5.** The predicted 3D structures and pocket binding sites of the ShPHT proteins. The putative 3D structures of ShPHT proteins were generated by the Phyre2 server, and the active catalytic binding sites were illustrated as red regions using the CASTp server 3.0.

### 3.7. GO and KEGG Analysis

To predict the function of ShPHT proteins, gene ontology (GO) annotation analysis was performed. A total of seventeen distinct functional groups were identified: nine ShPHT proteins are associated with biological processes (BF), five are involved in molecular functions (MF), and three are involved in cellular components (CC) (Table S4). In the MF processes, "inorganic phosphate transmembrane transporter activity and symporter activity" (GO: 005315 and GO: 0015293) are highly enriched GO terms. Similarly, among BP and CC, transmembrane transport (GO: 0055085), response to stimulus (GO: 0050896), and integral component of membrane (GO: 0016021) are the top GO terms. Additionally, in the KEGG analysis, most of the target genes were categorized into signaling and cellular process related to phosphate transport (Table S5).

### 3.8. Expression Analysis of PHT Genes under Salinity Stress

To investigate the putative physiological roles of *ShPHT* genes, we compared the expression between a salinity-tolerant wild genotype and a salinity-sensitive commercial sugarcane variety. A total of 23 *PHT* genes were investigated for their relative expression levels using qRT-PCR. In the salinity-sensitive genotype, only ShPHT1-4 and ShPHT2-1 showed relatively higher expression; *ShPHT1-1*, *ShPHT1-2*, *ShPHT1-3* genes were slightly upregulated and other genes showed either stable or slightly decreased expression. The *ShPHT1-1*, *ShPHT1-2*, and *ShPHT1-3* genes were predominantly expressed in both genotypes, though their expression was dramatically induced under salt stress (Figure 6; Table S6)). Furthermore, in the wild genotype all seven *ShPHT1* subfamily members exhibited differential expression, while ShPHT1-6 had no significant change under salinity treatment. In the ShPHO1 subfamily, ShPHO1-2, and ShPHO1-3 were upregulated in the wild genotype in comparison to the sensitive cultivar. These findings show that these upregulated genes may be involved in modulating sugarcane stress responses to salinity.

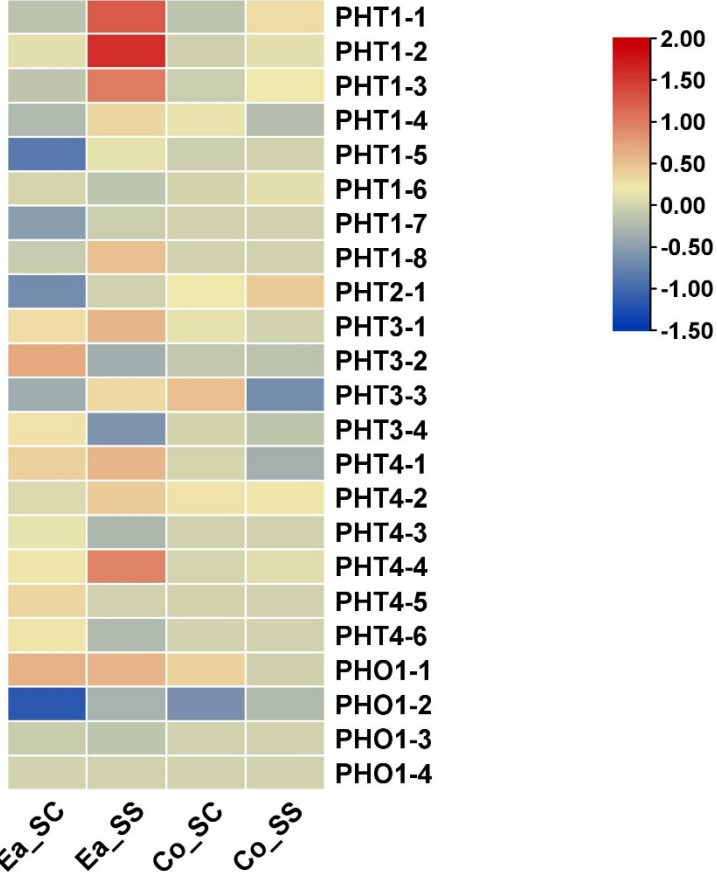

**Figure 6.** Heat map of *PHT* genes responding to salinity stress in sugarcane root based on the analysis of qRT_PCR expression of wild genotype *E. arundinaceus* IND99-907 (salinity tolerant) and *Saccharum* hybrid Co97010 (salinity sensitive) cultivars. Ea_SS represents the *E. arundinaceus* salinity stress, Ea_SS represents the *E. arundinaceus* salinity control, Sh_SS represents the *Saccharum* hybrid salinity-sensitive stress variety, and Sh_SC represents the *Saccharum* hybrid salinity-sensitive control variety. Red indicates high expression and green indicates low expression.

### 3.9. Putative cis-Regulatory Element Analysis

To further investigate the transcriptional mechanism of *ShPHTs*, the cis-acting regulating elements situated 2000 bp upstream from the transcription start site (TSS) of 23 *ShPHT* genes were analyzed. The results showed 147 types of 11490 cis-acting elements (Table S7). Several cis-acting regulatory elements related to cell and developmen-

tal elements, light-responsive elements, stress-responsive elements, hormone-responsive elements, and tissue/organ-specific elements were discovered in the promoter region (Figure 7A). We mainly characterized these into three categories. First, for the cis-elements related to P response (P1BS—PHR1-binding sites), the P1BS element was present in the all promoter regions of ShPHTs. ShPHT1-7 had 18 P1BS, while another gene in the ShPHT1 family ranged from 3 to 10 and other genes such as PHT2, PHT3, PHT4, and PHO had 2 to 9 of the P1BS elements. Elements related to abiotic stress were predicted in six categories, of which the ABRE (ABA-responsive element), MYB (drought responsive), DRE (dehydration-responsive), and GT-1 (salt-induced) components are relatively large (Figure 7B). Cis-elements related to tissue-specific expression are associated with, leaf expression (DOFCOREZM and GT1CONSENSUS), root expression (ACGTROOT1, OSE1ROOTNODULE, OSE2ROOTNODULE, LEAFYATAG, RAV1AAT, and RHERPAT-EXPA7), flower expression (AGAMOUSATCONSENSUS, AGL2ATCONSENSUS, CAR-GATCONSENSUS, and CARGATCONSENSUS), pollen expression (QELEMENTZMZM13 and PSREGIONZMZM13), and seed expression (PROXBBNNAPA, RYREPEAT4, RYRE-PEATLEGUMINBOX, SEF4MOTIFGM7S, CAATBOX1, and SPHZMC1). The presence of the majority of these cis-acting regulatory elements in *ShPHT* gene promoter demonstrates their importance in the regulation of the *PHT* gene family in phosphate transport, distribution, and tissue-specific expression under biotic/abiotic stress conditions (Table S7). Transient expression analysis using a GUS assay of EaPHT1; 2-GUS in sugarcane leaf bits (Figure 7C) provided rapid analysis of promoter efficiencies in expression of downstream Pi-responsive genes.

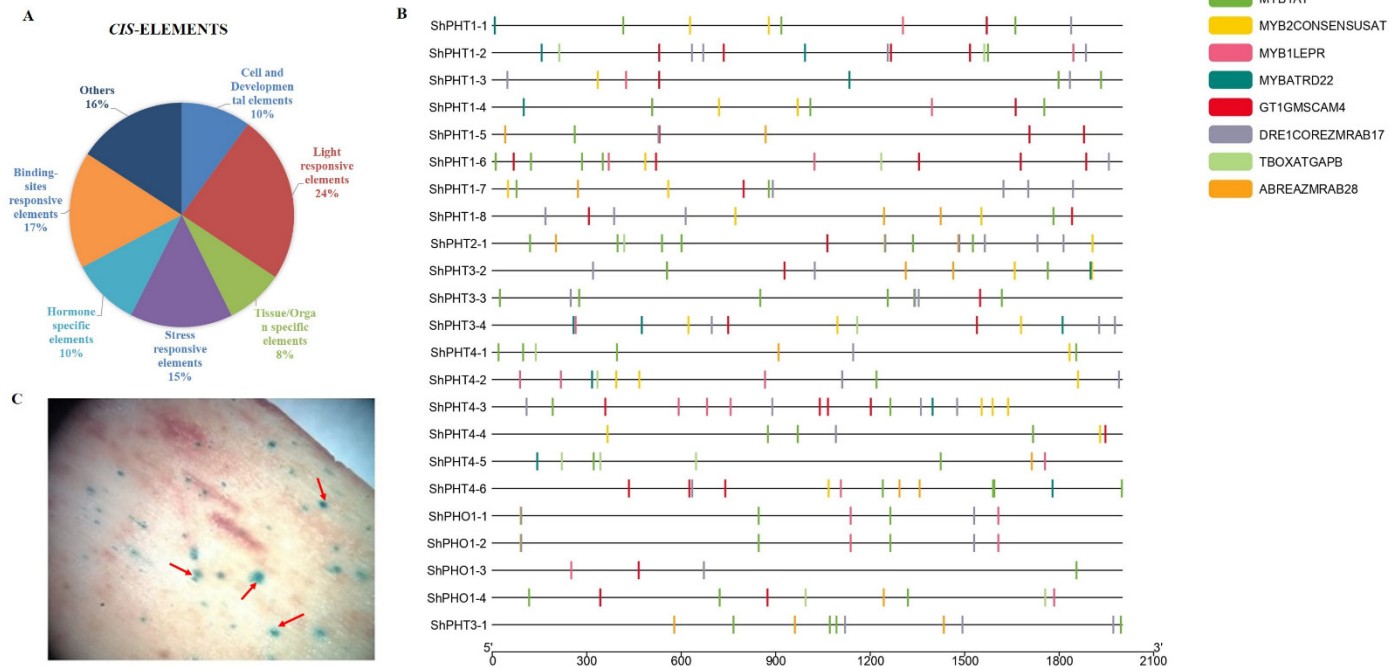

**Figure 7.** Analysis of the ShPHT promoter regions. (**A**) Distribution of cis-regulatory elements into promoter regions of *ShPHT* genes (2 kb upstream from the CDS region). Classification of identified regulatory elements based on function, such as cell, developmental, light, tissue/organ, stress, hormone, binding domain, and others. (**B**). Distribution of different types of stress-related cis-regulatory elements. (**C**). Transient expression analysis of EaPHT1; 2 promoter. Red arrows indicate the GUS Expression identified on sugarcane leaf pieces.

## 4. Discussion

Phosphate is an essential nutrient involved in plant growth and metabolism. In plants, *PHT* genes accomplish Pi uptake and translocation. To date, members of the PHT protein family have been reported in many plant species, including barley, rice, soybean, tomato,

foxtail millet, maize, poplar, potato, wheat, apple, sorghum, rapeseed, camelina, duckweed, capsicum, salt cress, and tea [7–9,41–55]. However, a detailed study of the sugarcane *PHT* gene family has not yet been reported. Sugarcane is an important commercial crop grown worldwide for sugar and ethanol production. Environmental stresses are the major limitation on crop productivity, and result in significant yield losses. In addition, sugarcane production is influenced by various unfavorable abiotic factors, namely, temperature, salt stress, nutritional deficits, and drought, which directly affect plant growth and culminate in shoot and root growth reduction [15]. The combination of salt and P deficiency is recognized to be most detrimental to plant health. One of the key ways to increase salt tolerance and production is to manage mineral nutrients in order to reduce salt-induced nutritional problems in plants. Less P availability, increased P fertilizer cost, and worse plant P usage efficiency all justify the need to investigate the phosphate transporter gene family in sugarcane [12].

Recent advances in sugarcane genome sequencing have made it possible to conduct a wide-ranging in silico analysis of the sugarcane *ShPHT* gene family in reference to *A. thaliana*, *Z. mays*, and *S. bicolor* genomes. In the present study, the sugarcane *Saccharum* spp. hybrid R570 genome was analyzed to identify the genes encoding PHTs. A total of 23 *ShPHT* genes were identified in the *Saccharum* spp. hybrid genome, including eight PHT1, one PHT2, four PHT3, six PHT4, and four PHO1 genes. Consistent with previous reports, the number of PHTs in sugarcane was less than that in the rice, poplar, apple, and sorghum genomes [8,42,46,49]. The phylogenetic relationship revealed that the prevalence of individuals in different subfamilies is quite similar in sugarcane, sorghum, and *Z. mays* (Figure 2). It has been proposed that proteins with greater homology within a class/subfamily may have comparable activities. To identify the sequence features of the putative *PHT* genes, the conserved domains of the PHT proteins were predicted using NCBI's conserved domain database. Consistent with previous studies, the conserved domains found in all the proteins were confirmed as typical phosphate transporter proteins (Figure 1).

PHT1 proteins are the largest and most widely studied subfamily of plant Pi transporters. Our comparative analysis of PHT1 proteins in sugarcane revealed eight ShPHT1 subfamilies, with a conserved core motif element GGDYPLSATIxSE, twelve transmembrane (TM) domains, and predicted localization on the plasma membrane, similar to the PHT1 transporters identified from other plant species [56]. Conversely, the smallest subfamily, PHT2, is a low affinity chloroplast Pi transporter mostly reported with one *PHT2* gene in many species, such as Arabidopsis, capsicum, duckweed, maize, potato, rice, wheat, Sorghum, and salt cress [57]; in very few species is it reported in more than one gene (two in poplar, tea and apple, three in Camelina and lupin, and two to four in Brassica genomes; in *Camellia sinensis*, no PHT2 subfamily was reported) [55]. The transmembrane domains of a protein play a crucial role in transporting various substances across the biological membranes. In sugarcane, the PHTs contain 0 to 13 transmembrane domains, similar to other crops. The *PHT2* gene showed thirteen TMDs located in the chloroplast, similar to Arabidopsis, Sorghum, and *Z. mays*. The four PHT3s and six ShPHT4s from sugarcane are mostly related to Sorghum and *A. thaliana*. The ShPHT3s are located on the mitochondria, and ShPHT4s are localized mainly in chloroplast and plasma membrane. In plants, the PHO1 subfamily is another important Pi-transporter that is essential for long-distance Pi transport from roots to shoots. In Arabidopsis, eleven *PHO1* genes (*PHO1* with ten homologous genes) were reported to be localized on endomembrane; comparatively, in sugarcane fewer genes (four *PHO1* genes) are predicted to be located on the plasma membrane [5].

The crystal structure of the plant PHT family is not yet present in the Protein Data Bank. In 2018, Kumar et al. [22] predicted the wheat PHT1 subfamily protein 3D structures with homology modelling, and recently the 3D structures of duckweed PHT proteins (PHT1 to PHT5 subfamilies) have been predicted [52]. The prediction of proteins' 3D structure and ligand-binding sites can provide valuable information about protein function. Based on our results, arginine, alanine, leucine, serine, valine, phenylalanine, glycine, threonine, isoleucine, and cysteine were frequently found in the pocket sites of all the candidate PHT

proteins, revealing their function and interaction during various environmental stimuli. GO and KEGG annotations of the *PHT* gene family describe contour features such as phosphate ion transport, symporter activity, inorganic anion transmembrane transport, response to stimulus, and cellular response to phosphate starvation, and further reveal the *ShPHT* genes encoding inorganic phosphate transporter activity functions.

*PHT1* genes have been studied in numerous plants and found to have a dominant expression in the root, indicating that these genes may play a role in Pi capture and absorption. Intriguingly, the maize *PHT1* subfamily genes *ZmPHT1:1*, *ZmPHT1:3*, *ZmPHT1:4*, *ZmPHT1:8*, and *ZmPHT1:9* were mostly expressed in the roots [9]. In Sorghum (*SbPHT1:2* and *SbPHT1:11*) and Arabidopsis (*AtPHT1.1* and *AtPHT1.4*), only two genes were highly expressed in roots [8,58]. In our study, in the salinity-tolerant *E. arundinaceus* wild genotype (IND 99-907) the genes *PHT 1-1*, *PHT 1-2*, and *PHT 1-3* were upregulated more than in the sugarcane salinity-sensitive genotype Co 97010. *PHT1-2* and *PHT1-3* showed higher expression in roots, demonstrating that they play a substantial role in Pi uptake from the soil and redistribution under salinity stress. The PHT2 subfamily that co-transports H+/Pi into chloroplasts might have low-affinity Pi transporters and is expressed preferentially in the shoots, especially in rosette leaves [57]. Expression of the PHT3, PHT4, and PHT5 subfamilies was widely reported in various tissues [49]. In sugarcane, the *PHT4-4*, *PHO1-1*, and *PHO1-2* genes are upregulated, while other genes remain the same or have decreased expression in roots. Gene expression is highly regulated by the *cis*-element present in the promoter region of the gene. The *cis*-elements in ShPHT promoters indicate that most of the genes contain multiple *cis*-acting elements related to stress, hormonal, and light-responsive *cis*-elements (Table S7). The *cis*-acting elements related to the responsiveness of abiotic/biotic stresses such as MYB and MYC are highly distributed in the upstream region of sugarcane *PHT* genes. Earlier reports have demonstrated the involvement of the MYB-type [59], WRKY-type [60,61], and bHLH-type [62] transcription factors in regulation of *PHT* genes. The phosphate starvation response (PHR) and PHR1-like (PHL) protein family belongs to the MYB transcription factors, and positively regulates *PHT* gene expression by binding to P1BS [63]. In addition, the light-responsive cis-elements play an important role in regulating the PHR1 protein under Pi starvation [64]. In accordance with this, our study found that the P1BS element was present in all the *ShPHT* gene promoters, and the number of PIBS sites ranged from 3 to 18. In vitro transient expression of the EaPHT1-2 promoter confirmed its activity, providing an indication that *PHT* genes and promoters have high potential to regulate the cellular processes associated with salt stress resistance.

## 5. Conclusions

The current work is the first to provide a comprehensive analysis of the sugarcane *PHT* gene family in terms of evolutionary connections, gene structures, conserved motifs, cis-acting elements, and their ontology and expression patterns. The study identified 23 *ShPHT*s genes, which were classified into five groups (*ShPHT1* to *ShPHT4* and *ShPHO1*) based on their phylogenetic relationships. The gene structure and conserved domains of each *PHTs* subfamily were extremely similar to that of their orthologues in *Sorghum*. Expression analysis of sugarcane root-specific *PHT* genes, namely, *ShPHT1-1*, *ShPHT1-2*, and *ShPHT1-3*, shows higher expression in the wild genotype IND 99-907 (salinity-tolerant) compared to the commercial variety Co 97010 (salinity-sensitive) under salt stress conditions. Analysis of cis-acting elements of ShPHT promoters showed the differential distribution of multiple abiotic stress elements; specifically, P1BS in the promoter region and the transient expression of GUS driven by the EaPHT1-2 promoter in sugarcane revealed potential promoter activity. The insights provided by this study can help to decipher the genetic information of the *PHT* genes and their potential use in the generation of improved sugarcane varieties adapted to Pi starvation and other salinity stresses for sustainable production. Nevertheless, additional investigation is required to confirm the purposeful role of PHT gene promoters under abiotic and Pi-induced stress and their response with respect to other stress conditions.

**Supplementary Materials:** The following supporting information can be downloaded at: https://www.mdpi.com/article/10.3390/su142315893/s1, Figure S1: Prediction of subcellular localization of ShPHT proteins. The *x*-axis represents the names of the ShPHT proteins and the *y*-axis represents the percentage of localization. Abbreviations: Plas—Plasma membrane; cyto: cytosol; Extra–extracellular; Pero–peroxisome; Chlo—Chloroplast; E.R.—Endoplasmic reticulum; Vacu—Vacuole; Cyto_plas-cytosol and plasma membrane; Mito—Mitochondria; cyto_nucl: cytosol and nucleus; Nucl—Nucleus; Figure S2: Prediction of posttranslational phosphorylation and glycosylation modification of ShPHT proteins; Figure S3: Transmembrane domains (TMDs) prediction of 23 ShPHT proteins by TMHMM (transmembrane prediction using Hidden Markov Models) server; Table S1: The structural features of motif 1–20; Table S2: Secondary structure of amino acid sequences in ShPHT by SOPMA; Table S3: Phosphorylation and glycosylation sites in ShPHT proteins; Table S4: Functional annotation of ShPHT proteins based on PANNZER2 output; Table S5: KEGG analysis of PHT proteins assigned by the BlastKOALA tool; Table S6: The RNA-Seq data of the PHT genes; Table S7: The cis-elements in promoter sequences of Phosphate transporter genes in Sugarcane.

**Author Contributions:** Conceptualization, N.M. and A.C.; Formal analysis, N.M. and V.P.; Investigation, N.M.; Methodology, V.P. and M.C.; Supervision, A.C.; Validation, N.M.; Writing—original draft, N.M.; Writing—review and editing, M.C., V.R., M.R., H.G. and A.C. All authors have read and agreed to the published version of the manuscript.

**Funding:** This research received no external funding.

**Institutional Review Board Statement:** Not applicable.

**Informed Consent Statement:** Not applicable.

**Data Availability Statement:** The data used in this study are available in the NCBI database under BioProject accession number PRJNA716503.

**Acknowledgments:** We wish to thank all the reviewers and editors for their careful reading and helpful comments on this manuscript. We gratefully acknowledge Indian Council of Agricultural Research (ICAR)-Sugarcane Breeding Institute (SBI) for supporting the current research. The authors also thank Anilkumar C, Scientist, NRRI, Cuttack for his help in editing the manuscript.

**Conflicts of Interest:** The authors declare that there is no conflict of interest regarding the publication of this paper.

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
