# Peer review of "Genome-Wide In Silico Identification, Structural Analysis, Promoter Analysis, and Expression Profiling of PHT Gene Family in Sugarcane Root under Salinity Stress"

_sustainability, doi:10.3390/su142315893_

Round 1

Reviewer 1 Report

The manuscript by Murugan  et al tried to take an integrative investigation of PHT gene family in Sugarcane. However, the methods and theoretical innovations are not sufficient due to the PHT gene family has been reported in many species like apple, poplar, and green algea and so on. The authors should refine some of their analysis. Below I outline several major and minor issues.

Major revisions:
1. My key issue is expression analysis (Figure 6). First, the findings are purely based on transcriptome data. It is well known that biological replicates are required for transcriptomic investigation of protein expression in various plant tissues. Unfortunately, none were found in this study. Second, the expression data process is unclear, and it seems not do scale normalization in fig6. Last, quantitative RT PCR is strongly recommended for experimental confirmation of expression patterns. Alternatively, the expression data in Figure 5 should be omitted entirely.

2. Figure 4A. ShPHT4-2 has unusually long introns. This raises doubts about the functionality of this gene. Can the authors exclude the possibility of the gene represent pseudogene?

3. I wonder how significant the differences in Ka/Ks values are.

4.The meaning of analysis results should be elucidate (please refer to https://doi.org/10.3389/fpls.2019.00565)

Minor revisions:

1. In line 105, were the protein, CDS, and gff3 annotation files downloaded from the same source as DNA sequence?

2. To better show the relevant classification results, five clades should be labelled in figure 2.

3. The ShPHT4-2 was out-of-order in figure 4B.

4. Please check the English grammar in this manuscript, for example,

line 203, ‘backend database change into  background database’.

line 280, ‘resulting that the motifs being conserved domain sequences change into resulting that the motifs were conserved domain sequences. 

Author Response

Manuscript: sustainability-1886282

Title: Genome-Wide In-Silico Identification, Structural Analysis, Promoter Analysis, and Expression Profiling of PHT Gene Family in Sugarcane Root Under Salinity Stress

September 14th, 2022

Here is a point-by-point response to the reviewers’ comments and concerns.

Comments from Reviewer 1

Major revisions

Comment 1: My key issue is expression analysis (Figure 6). First, the findings are purely based on transcriptome data. It is well known that biological replicates are required for transcriptomic investigation of protein expression in various plant tissues. Unfortunately, none were found in this study. Second, the expression data process is unclear, and it seems not do scale normalization in fig6. Last, quantitative RT PCR is strongly recommended for experimental confirmation of expression patterns. Alternatively, the expression data in Figure 5 should be omitted entirely.

Answer: We added data for quantitative real time-PCR and which can be seen in the figure 6 and also rewrote material and methods section. Modifications made in the revised manuscript is highlighted with yellow colour.

Comment 2: Figure 4A. ShPHT4-2 has unusually long introns. This raises doubts about the functionality of this gene. Can the authors exclude the possibility of the gene represent pseudogene?

Answer: Thank you and this point is very important. However, we believe that the ShPHT4-2 may not be a pseudogene, because in the Sorghum PHT study, among all the SbPHT genes, SbPHT4-2 has the longest intron structure (For your kind reference; https://www.ncbi.nlm.nih.gov/pmc/articles/PMC6921035/). This shows a similar intron structure between the PHT4-2 of Sorghum and Sugarcane. Our own experience with other genes too, long intron with exceptional functional role.

Comment 3: I wonder how significant the differences in Ka/Ks values are

Answer: The Ka/Ks ratio is seen as an important indicator of selective constraints in gene diversification. The duplicated genes with a high Ka/Ks ratio (>1) are deemed to be evolving under positive selection, Ka/Ks = 1 indicates neutral selection, and Ka/Ks < 1 indicates purifying selection. The ratios for seven duplicated ShPHT gene pairs were calculated and they were all less than 0.5 (Table 2), which implied that the PHT genes from sugarcane had mainly undergone strong purifying selection after the duplication events with limited functional divergence. The evolution in duplicated gene pairs has been done during recent and ancient times as well. Such findings will help for future studies on PHT genes in sugarcane related to the functional and evolutionary characterization.

Comment 4: The meaning of analysis results should be elucidate (please refer to https://doi.org/10.3389/fpls.2019.00565)

Answer: Agree. Thank you for this suggestion. Accordingly, we have revised the manuscript to emphasize this point. Please check the revised manuscript.

Minor revisions:

Comment 1: In line 105, were the protein, CDS, and gff3 annotation files downloaded from the same source as DNA sequence?

Answer: Thank you! Yes, the complete sequence dataset including the protein, CDS, gff annotation files and DNA sequences were downloaded from the sugarcane genome hub database.

Comment 2: To better show the relevant classification results, five clades should be labelled in figure 2.

Answer: Thank you! As suggested by the reviewer, we have made the changes in Figure 2-the phylogenetic tress and included in the revised manuscript.

Comment 3: The ShPHT4-2 was out-of-order in figure 4B.

Answer: While we appreciate the reviewer’s feedback, we respectfully disagree. We think the gene PHT4-2 was not in out-of-order, other genes like PHT 4-5, and PHT 4-6 of the same sub-family shows similar motif structure.

Comment 4: Please check the English grammar in this manuscript, for example,

line 203, ‘backend database’ change into  ‘background database’.

line 280, ‘resulting that the motifs being conserved domain sequences’  change into ‘resulting that the motifs were conserved domain sequences’. 

Answer: Thank you for pointing the mistakes. The sentences have been corrected at the reviewers’ suggestion.

Reviewer 2 Report

The manuscript by Murugan et al. represents a comprehensive analysis of the PHT phosphate transporter family in sugar cane. The authors have identified the members of the family and analyzed their structure, domain organization, chromosomal location and sequence relationship with other PHT proteins in monocots and a dicot species (Arabidopsis). Attention was paid to the transcriptional regulation of the family, including a transcriptome analysis for salt stress response and an in silico analysis of upstream sequences for the presence of cis-elements.

Below, my comments and suggestions:

The TMHMM server v.2.0 has returned no transmembrane domains for five PHT3 proteins. It will be important to comment why this is so, and whether the authors are confident that the full-length sequences have been recovered during the initial search.

To the phylogenetic analysis, the authors state that “It is interesting to note that the proteins of PHT1 are closely related among monocot species rather than A. thaliana, suggesting that they diverged from a common ancestor.”

Interestingly, this is to a certain extent true for the AtPHO and some ZmPHT2/3 proteins as well. Is there any data, expression or functional that would support this divergence? Would be helpful for the reader if authors discuss this point in the text.

The authors need to better argument why salinity stress condition was used to evaluate the expression of the phosphate transporter family. A potential line of argumentation would be that phosphate accumulating Arabidopsis mutants seem more salt stress tolerant than the wild types (Miura et al. 2011 Planta)

Further discussion is needed on the cis-element analysis, especially in the sense of the role of light. Figure 7a shows that almost 25% of the identified promoter elements are related to light responses. This supports a wave of recent findings that light responses and light-responsive transcription factors, such as PIF4, HY5 and the NF-Y complex function as the coordinators of nutrient homeostasis (Brumbarova and Ivanov 2019 iScience). In addition, PHR1 itself seems to be light-regulated to orchestrate phosphate starvation response (Liu et al. 2017 Plant Cell).

Figures 3 and 4: the letters are too small for reading and need to be enlarged.

Line 355: both the terms “genotype” and “variety“ are used, further in the same paragraph, “wild type” and later “wild genotype” and “wild-type”. Please, select which terms to use and do it consistently.

Author Response

Manuscript: sustainability-1886282

Title: Genome-Wide In-Silico Identification, Structural Analysis, Promoter Analysis, and Expression Profiling of PHT Gene Family in Sugarcane Root Under Salinity Stress

September 14th, 2022

Here is a point-by-point response to the reviewers’ comments and concerns.

Comments from Reviewer 2

Comment 1: The TMHMM server v.2.0 has returned no transmembrane domains for five PHT3 proteins. It will be important to comment why this is so, and whether the authors are confident that the full-length sequences have been recovered during the initial search.

Answer: Thank you for this suggestion. It would have been interesting to explore this aspect. However, in our study, the full length of the gene was extracted and further confirmed by the above mentioned primary sequence analysis methods. Thus, this would not be a possible reason that the TMHMM server v.2.0 has returned no transmembrane domains for all the genes in the specific subfamily.  Further, the PHT3 subfamily in Sorghum also has no transmembrane domains.

Comment 2: To the phylogenetic analysis, the authors state that “It is interesting to note that the proteins of PHT1 are closely related among monocot species rather than A. thaliana, suggesting that they diverged from a common ancestor.”

Answer: Agree. Members of the PHT1 subfamily (Figure 2) are grouped into two main sub-clades monocotyledon (Zea mays, Sorghum and Sugarcane), and dicotyledon (A. thaliana), suggesting that the common ancestor of PHT1 proteins in this group arose before the division of monocotyledon and dicotyledon plants.

Comment 3: Interestingly, this is to a certain extent true for the AtPHO and some ZmPHT2/3 proteins as well. Is there any data, expression or functional that would support this divergence? Would be helpful for the reader if authors discuss this point in the text.

Answer: We thank the reviewer for the suggestion, however, to the best of our knowledge, there is no evidence related to this divergence

Comment 4: The authors need to better argument why salinity stress condition was used to evaluate the expression of the phosphate transporter family. A potential line of argumentation would be that phosphate accumulating Arabidopsis mutants seem more salt stress tolerant than the wild types (Miura et al. 2011 Planta)

Answer: As suggested by the reviewer, we have revised the manuscript. Please check the revised version.

Comment 5: Further discussion is needed on the cis-element analysis, especially in the sense of the role of light. Figure 7a shows that almost 25% of the identified promoter elements are related to light responses. This supports a wave of recent findings that light responses and light-responsive transcription factors, such as PIF4, HY5 and the NF-Y complex function as the coordinators of nutrient homeostasis (Brumbarova and Ivanov 2019 iScience). In addition, PHR1 itself seems to be light-regulated to orchestrate phosphate starvation response (Liu et al. 2017 Plant Cell).

Answer: We thank the reviewer for the suggestion. We have added the suggested content to the manuscript. The change can be found in the revised manuscript.

Comment 6: Figures 3 and 4: the letters are too small for reading and need to be enlarged.

Answer: Thank you for the suggestion. We have revised the figures and included them in the revised manuscript. Please check the revised version.

Comment 7: Line 355: both the terms “genotype” and “variety” are used, further in the same paragraph, “wild type” and later “wild genotype” and “wild-type”. Please, select which terms to use and do it consistently.

Answer: Thank you! The sentence is reframed as per the reviewer suggestion.
